# Investigation of Molecular Mechanism of Cobalt Porphyrin Catalyzed CO_2_ Electrochemical Reduction in Ionic Liquid by In-Situ SERS

**DOI:** 10.3390/molecules28062747

**Published:** 2023-03-18

**Authors:** Feng Wu, Fengshuo Jiang, Jiahao Yang, Weiyan Dai, Donghui Lan, Jing Shen, Zhengjun Fang

**Affiliations:** Hunan Provincial Key Laboratory of Environmental Catalysis & Waste Recycling, School of Material and Chemical Engineering, Hunan Institute of Engineering, Xiangtan 411104, China

**Keywords:** SERS, cobalt protoporphyrin, pholorin, ionic liquid

## Abstract

This study explores the electrochemical reduction in CO_2_ using room temperature ionic liquids as solvents or electrolytes, which can minimize the environmental impact of CO_2_ emissions. To design effective CO_2_ electrochemical systems, it is crucial to identify intermediate surface species and reaction products in situ. The study investigates the electrochemical reduction in CO_2_ using a cobalt porphyrin molecular immobilized electrode in 1-*n*-butyl-3-methyl imidazolium tetrafluoroborate (BMI.BF4) room temperature ionic liquids, through in-situ surface-enhanced Raman spectroscopy (SERS) and electrochemical technique. The results show that the highest faradaic efficiency of CO produced from the electrochemical reduction in CO_2_ can reach 98%. With the potential getting more negative, the faradaic efficiency of CO decreases while H_2_ is produced as a competitive product. Besides, water protonates porphyrin macrocycle, producing pholorin as the key intermediate for the hydrogen evolution reaction, leading to the out-of-plane mode of the porphyrin molecule. Absorption of CO_2_ by the ionic liquids leads to the formation of BMI·CO_2_ adduct in BMI·BF_4_ solution, causing vibration modes at 1100, 1457, and 1509 cm^−1^. However, the key intermediate of ${\mathrm{CO}}_{2}^{-}\cdot$ radical is not observed. The *υ*(CO) stretching mode of absorbed CO is affected by the electrochemical Stark effect, typical of CO chemisorbed on a top site.

## 1. Introduction

Carbon dioxide is a harmful by-product of fossil fuels combustion that poses serious environmental problems such as global warming as its concentration in the atmosphere accumulation. However, it is also an abundant raw material that can be converted into high-value products, such as CO [1], formic acid [2], and methanol [3] by means of electrochemical reduction [4,5,6]. This process not only addresses environmental issues, but also provides a sustainable energy source. Unfortunately, CO_2_ is an inert, linear molecule that requires high energy input to activate and form the common first key intermediate ${\mathrm{CO}}_{2}^{-}\cdot$ anion radical which requires a high-energy potential with an equilibrium potential of −1.9 V vs. a standard hydrogen electrode (SHE, pH = 7) [7]. As a result, electrochemical reduction in CO_2_ must proceed catalytically, with sensitivity of the products determined by the electrode material used. Early investigations into CO_2_ electrochemical reduction were generally conducted in aqueous solution, resulting in a primary reaction pathway competition from the hydrogen evolution reaction (HER). Additionally, the limited solubility of CO_2_ in aqueous solution results in low faradaic efficiency for the products. To overcome these challenges, organic solvents and room-temperature ionic liquids (RTILs) have been implemented as solvents for the reaction [8,9,10].

Room-temperature ionic liquids are a unique class of solvents that are being investigated for their potential as superior alternatives to organic solvents in electrochemical applications. This is because they have a wide electrochemical window [11], thermal and chemical stability [12], negligible volatility [13,14], and can be used as electron transfer mediators for redox catalysis [14]. In recent years, RTILS have gained attention as promising materials for electrochemical reduction CO_2_ technology due to their high selectivity and relatively high CO_2_ adsorption capacity. The solubility of CO_2_ in a serious of fluorine-free RTILs has been investigated by Palgunadi et al. [15,16]. They suggested that the longer alkyl chain in the dibutylphosphate anion creates a larger free volume to accommodate more CO_2_ molecules. Azhar et al. investigated the solubility of CO_2_ in an aqueous solution of monoehanolamin and 1–butyl–3–methylimidazolium dibutylphosphate (BMIM·DBP) ionic liquid hybrid solvents [17]. Based on response surface methodology (RSM) and mathematical modelling, they predicted the optimum conditions for CO_2_ absorption.

Rosen et al. reported 2011 that the overpotential of the electrochemical CO_2_ reduction to CO is below 0.2 volt by utilizing EMIM–BF_4_ ionic liquid to lower the activation energy of CO_2_. The ionic liquids also helped increases the selectivity of CO_2_ to CO to above 95%. Ref. [18] Latter, Zhao et al. investigated the role of cation of ionic liquids in the heterogeneous/homogeneous CO_2_ electrocatalysis reduction process. According to their assertion, the presence of imidazolium and pyrrolidium cations significantly improved the electroreduction in CO_2_ using Ag as an electrocatalyst, particularly in the presence of water [19]. The local ionic liquid environment has been established at an iron porphyrin catalyst to enhance electrocatalytic performance of CO_2_ to CO reduction by Khadhraoui et al. [5]. Despite the fact that ionic liquids possess the ability to stabilize the potential intermediate ${\mathrm{CO}}_{2}^{-}\cdot$, rendering them highly desirable for the CO_2_ electrochemical reduction technique, the implementation of the technology utilizing RTILs is still a long way off [18]. It is essential to have a comprehensive understanding of the fundamental mechanisms of electrochemical reduction in ionic liquids for the rational design of such systems. As a result, considerable research efforts have been directed towards this area.

Tanner et al. conducted a screening experiment involving a combinations of metal electrodes, RTIL cations and anions to unravel the fundamental mechanism involved in electrochemical CO_2_ reduction [20]. Their findings revealed that Ag was the most efficient electrode material in [Bmim][NTf_2_] and the CO_2_ reduction activity was dependent on cations. They suggested that components of ionic liquids played a complex role in the CO_2_ reduction mechanism by potentially modifying the nature of the double-layer formed at the electrode surface. Lau et al. conducted a screening experiment that yielded intriguing results. Their findings indicated that the improved activity of imidazolium based RTILs was attributed to the hydrogen atoms at the C4 and C5 positions, rather than those at the C2 position [21]. Theoretical investigations have also been extensively explored the fundamental mechanism of electrochemical CO_2_ reduction in RTIL-based solvents. Norskov’s group, in particular, has focused on analyzing the impact of the electric field generated by cations in the vicinity of the electrode surface [22]. Additionally, Urushihara et al. conducted first-principles-based thermodynamic stability analysis (Pourbaix diagram) of [Emim] at the Ag(111) and water interface, which varied as a function of electrochemical potential and [Emim] concentration. Their simulations involved the solvation effect, which was modeling using a combination of explicit water molecules adsorbed on the electrode surface and implicit water modeled as a dielectric medium [23]. Despite the extensive research efforts, the comprehensive understanding of the mechanism of RTIL-based electrochemical CO_2_ reduction remains limited.

In-situ surface-sensitive techniques are essential for understanding the structure-activity relationship and reaction mechanism at a molecular level in heterogeneous catalysis. In the case of Pt electrode and [Emim][BF_4_] system, Rosen et al. utilized the SFG (Sum Frequency Generation) spectrum observed CH_3_ bending and ring stretching modes at 1430 cm^−1^ and 1570 cm^−1^, respectively. These observations suggest that the [Emim] cations preferentially reside at the electrode surface during electrolysis. Furthermore, they found that during the electrochemical CO_2_ reduction reaction, as the cathode potential became more negative, a sharp peak at 2348 cm^−1^ appeared gradually attributing to the formation of an [Emim] –CO_2_– [BF_4_] complex at the electrode surface [24]. Osawa’s group has used surface-enhanced infrared absorption spectroscopy (SEIRAS) to investigate the interface between an Au electrode and either pure 1-butyl-3-methylimidazolium bis(trifluoromethanesulfonyl)amide ([Bmim][TFSA]) or a [Bmim][TFSA]/water mixture solvent. By directly integrating specific SEIRAS peaks during cyclic voltammetry measurements, the authors were able to observe potential-dependent restructuring of the solvent components at the electrode surface. This restructuring included changes in the orientation and interactions of the [Bmim][TFSA] cations and anions and water molecules [25,26].

The other widely used in-situ techniques for this purpose is surface-enhanced Raman spectroscopy (SERS), which provides rich structural information on structural information on adsorbed molecules at very low concentration [27]. Santos et al. investigated the vibration of the RTILs derived from 1–n–butyl–3–methylimidazolium hexafluorophosphate (BMI·PF_6_) adsorbed on a silver electrode by SERS. The researchers observed that the BMI·PF_6_ cation adsorbs on the silver electrode for potentials more negative than −0.4 V vs. a Pt quasireference electrode (PQRE). At potentials more negative than −1.0 V, some imidazolium ring vibrational modes and N-CH_3_ vibrations were enhanced, suggesting that the imidazolium ring is parallel to the surface [28]. The electrochemical reduction in CO_2_ over a copper electrode in the RTILs, 1–n–butyl–3–methyl imidazolium tetrafluoroborate (BMI·BF_4_), was investigated using in-situ SERS by Santos Jr. et al. The SERS results showed that CO adsorbs on copper surface at two different surface sites. Additionally, a vibration at 2275 cm^−1^ was observed in the SERS spectra, confirming the presence of chemically adsorbed CO_2_. This suggests that the reduction in CO_2_ over the copper electrode in BMI·BF_4_ involves multiple surface reaction pathways [29]. Metalloporphyrins are known to exhibit excellent electrocatalytic activity towards CO_2_ when self-assembled or immobilized on graphite, glassy carbon and metal surfaces, but a plausible mechanism for their electrochemical reduction in CO_2_ was lacking solid evidence [30,31,32]. To address this gap, the combination of in-situ SERS and electrochemistry is used to interpret the reliable adsorbate structure with the precise controlling potential of working electrode (WE). Overall, in-situ surface-sensitive techniques such as SERS play a critical role in advancing our understanding of heterogeneous catalysis and developing more sustainable and efficient energy conversion technologies.

In this study, we utilize the sophisticated in-situ SERS methodology and electrochemical technique to authenticate the configuration of the intermediate species involved in the electrochemical reduction in CO_2_ facilitated by cobalt protoporphyrin in RTILs. Through analysis of the in-situ spectrum, we are able to infer the mechanism underlying the electrochemical reduction in CO_2_. This approach allows us to gain a deeper understanding of the electrochemical reduction process and the intermediate species involved. This study highlights the potential of in-situ SERS methodology to provide important insights into the mechanism of electrochemical reactions and to guide the development of more sustainable energy conversion technologies.

## 2. Results and Discussion

### 2.1. Characterization of Cobalt Protoporphyrin in ILs

The successful immobilization of cobalt protoporphyrin on carbon paper has been investigated using cyclic voltammetry (CV) as shown in Figure 1. During the potential range from 0 V to −2.0 V, the blank carbon paper presents a typical double layer voltammogram indicating its electrochemical stability in investigated potential range. As immobilized with cobalt protoporphyrin, there are two pair obvious redox peaks at −0.18 V and −1.48 V (vs. Ag/Ag^+^) assigned to electrochemically reversible redox processes of Co^III^/Co^II^ and Co^II^/Co^I^, respectively.

In Figure 2, the SERS spectra of a nanostructured Au working electrode modified with cobalt protoporphyrin in a N_2_–aturated room temperature ionic liquid are compared at different applied potentials. The characteristic vibrations of cobalt protoporphyrin are predominantly located in the range of 200–1700 cm^−1^ and are analyzed within this specific range. The applied potentials range from open circuit potential (OCP) to −1.5 V, with a measurement interval of 0.1 V, while the spectra are presented with a 0.2 V interval to prevent any overlap of spectroscopic signals. In Figure 2a, SERS spectra are presented for adsorbed cobalt protoporphyrin in the band range of 200–1000 cm^−1^. Previous literature has shown that specific vibrational modes of the porphyrin ring of hemin can undergo frequency shift based on the oxidation state, axial ligation, or coordination, and spin state of the central iron atom [33,34,35,36,37]. Typically, the *ν*(C–C) and *ν*(C–N) stretch vibrations of the porphyrin ring shift in frequency according to the iron oxidation state and can be found in the marker band range of 1300–1700 cm^−1^. In the case of hemin, the *ν*_4_ vibration mode corresponds to C–N stretch vibrations of the pyrrole subunits, which are sensitive to electron transfer in the π* orbital of the porphyrin ring and serve as an indicator of the iron oxidation state [38]. The typical *ν*_4_ vibration mode for ferric (Fe^3+^) and ferrous (Fe^2+^) hemin can be observed in the frequency range of 1368–1377 cm^−1^ and 1344–1364 cm^−1,^ respectively [39,40]. The *ν*_2_ marker vibration mode corresponding to the ν(C–C) stretch vibration of porphyrin ring, serves as another indicator of the coordination and spin state of the Fe^2+^/Fe^3+^ ions in hemin [41]. The bands at 1570 and 1574 cm^−1^ are mostly related to five-coordinated ferric hemin with high spin state, while bands at 1619 and 1607 cm^−1^ are likely correlated with ferrous hemin [42].

Form Figure 2b, a frequency at 1399 cm^−1^ is detected, which is assigned as the *ν*_4_ mode for Co^III^ at OCP. As the potential is scanned to the anodic direction at about −0.6 V, a vibration mode appears at a frequency of 1351 cm^−1^, indicating the partial formation of Co^II^. The observation is consistent with the cyclic voltammogram presented in Figure 1 where the reduction in Co^III^ happens at −2.5 V. Vibration modes above 1480 cm^−1^ are activated by the asymmetric disposition of the vinyl substituents as well as the vinyl C=C stretch and also dependent on the size of the porphyrin core. Frequencies related to C=C stretch are positioned at 1532, 1558 and 1642 cm^−1^, which hardly change as the potential is scanned to anodic direction, except 1642 cm^−1^, which starts disappearing at about −0.2 V. The features at 1123 and 1247 cm^−1^ are related to Raman active vibrations characteristic of the BMI^+^. The peak at 751 cm^−1^ is due to the ${\mathrm{BF}}_{4}^{-1}\;$ breathing mode. In the low-energy region, the intensity at 268 cm^−1^ is disappeared which is assigned to the Ag–Ag stretching model in the Ag substrate as the applied potential becomes more negative than −0.4 V [28]. However, the peak at 340 cm^−1^, which has not been identified yet, slightly gains intensity at potential −0.6 V.

### 2.2. Influence of Water as Proton Source

It has been previously established that the electrochemical reduction in CO_2_ requires protons; therefore, the sufficiency of water as a proton source in the ionic liquid electrolyte was investigated [43,44]. In Figure 3, the SERS spectrums obtained in the presence of an ionic liquid and water mixture saturated with N_2_, are presented. As the potential is scanned towards to negative direction, a notable increasing in the intensity of the peaks at 340, 611, 671 and 944 cm^−1^ is observed in the spectral range of 200 to1000 cm^−1^. These intensity changes have been attributed to an out-of-plane (oop) mode, which is a specific vibrational mode of the pronated porphyrin molecule, except for the peak at 944 cm^−1^ which corresponds to the in-plane mode. The Raman spectra of protonated porphyrin were simulated using DFT methods and compared with the experimental data by Gorski et al. [45]. Based on their observations, the peak at 340 cm^−1^ corresponds to the symmetrical tilting oop motion of the meso-carbon atoms and the nitrogen atoms of the pyrrolic rings, while the peak at 611 cm^−1^ is attributed to a twisting oop motion of the four pyrrolic rings with oop pyrrole ring deformation. The peak at 671 cm^−1^ is the result of an envelope oop motion of the four pyrrolic rings.

Metal porphyrins have shown great potential as molecular catalysts for the electrochemically triggered hydrogen evolution reaction (HER) due to their ability to participate in different proton-coupled electron transfer process (PCET) in the past few decades [46,47,48]. These processes generate intermediates that can donate hydrides to free protons, thus releasing hydrogen. The exact pathway of metalloporphyrin catalyzed HER is still a subject of ongoing research and debate, and several proposed mechanisms exist. The theoretical studies by Nocera and Hammes-Schiffer suggest that cobalt meso-tetrakis(pentafluorophenyl) porphyrin can act as a catalyst for the hydrogen evolution reaction (HER) by facilitating the reduction and protonation of the porphyrin macrocycle [49]. Hung and collaborators reported that the HER catalytic mechanism occurs via [P-Co^III^-H] or [P-Co^II^-H]^−1^ metal hydrides for cobalt porphyrins [50]. In the current study, the spectra collected suggest that the protonation of the porphyrin macrocycle may be the key process for the HER catalyzed by cobalt porphyrin.

In the spectra range of 1000–1600 cm^−1^, the features observed for the BMI^+^ in the pure ionic liquid solution are observed at the same wavenumber, 1126 and 1246 cm^−1^, apart from intensity. The *ν*(C-C) and *ν*(C-N) stretch vibrations of the porphyrin ring, which are indicative of metal oxidation state, also occurred in the marker band range of 1300–1700 cm^−1^. The peak at 1394 cm^−1^ is related to the *ν*_4_ mode of Co^III^ due to C–N stretch vibration of the pyrrole subunit, and the feature of C=C stretch is presented at 1566 cm^−1^ at open circuit potential. With decreasing potential, the peaks at 1348 and 1526 cm^−1^ emerge at –0.8 V, indicating the reduction in metal center from Co^III^ to Co^II^, which is consistent with the CV results presented in Figure 1. In the spectra range over 1600 cm^−1^, the only visible peak is at 2106 cm^−1^ with tiny intensity while applied potential is more negative than −0.4 V, which is likely assigned to the anti-symmetric stretching of the adsorbed CO_2_. This feature is probably due to the pre-existing dissolved CO_2_ in the ionic liquid electrolyte. Research by Wu et al. and Feroci et al. suggests that the BMI-carbene reacts with CO_2_ to form a BMI-CO_2_ carboxylate adduct [51,52], and the 2106 cm^−1^ feature observed is likely to be this type of adduct.

### 2.3. Characterization of CO_2_ Electrochemical Reduction

The CO_2_ electrochemical reduction activity of the cobalt protoporphyrin in 1–n–butyl–3–methyl imidazolium tetrafluoroborate and water (V:V = 9:1) solution is evaluated using linear sweep voltammetry (LSV), as shown in Figure 4a. The current density of cobalt protoporphyrin modified electrode is much higher than that of blank carbon paper electrode under a CO_2_ atmosphere (1 atm), suggesting efficient CO_2_ catalysis by cobalt protoporphyrin. On the other hand, the onset potential of cobalt protoporphyrin modified electrode is about 200 mV more positive than that of the carbon paper electrode, indicating the decreasing of activation energy by cobalt protoporphyrin. The gaseous products from electrochemical reduction in CO_2_ have been analyzed by gas chromatography (GC), as shown in Figure 4b. The CO_2_-to-CO faradaic efficiency of cobalt protoporphyrin keeps well above 90% in the applied potential window of −1.1 V to −1.5 V (vs. Ag/Ag^+^). With the potential becoming more negative, the faradaic efficiency of CO slowly deceases from 98% to 89%. At the same time, the faradaic efficiency of H_2_ gradually increases from 3% to 10%. The results indicate that the H_2_ evolution reaction will become competitive reaction at more negative potential.

The aim of the study was to gain insight into the mechanism of CO_2_ electrochemical reduction catalyzed by cobalt porphyrin in ionic liquid. To achieve this, SERS spectra are recorded for cobalt porphyrin immobilized electrode in CO_2_ saturated ionic liquid and water (V:V = 9:1) as presented in Figure 5. In the frequency range of 200–1000 cm^−1^, the spectra show little difference compared to those obtained with N_2_ saturated solution, with peaks observed at 342, 607, 678, 750 and 944 cm^−1^. The only noticeable difference is the initial potentials of peaks at 342, 607 and 678 cm^−1^ are more positive which is at −0.2 V. As elucidated above, those peaks are related to the out-of-plane mode due to the protonation of the porphyrin macrocycle.

The emergence of the peaks at 1100, 1457 and 1509 cm^−1^ in the spectra range of 1000–1600 cm^−1^ indicates the formation of BMI-CO_2_ adduct in the BMI·BF_4_ solution electrolyzed in the presence of CO_2_. It is noticeable that the initial potential for BMI-CO_2_ adduct is at −0.6 V which is more negative than that for the protonation of porphyrin macrocycle. The interaction between CO_2_ and ILs is possible due to the capability of ILs to encapsulate CO_2_ within cavities close to alkyl groups and aromatic protons of the IL [53]. This interaction does not interfere with the interaction of the counter ion of the IL, which result in the formation of the BMI-CO_2_ adduct. Some ILs, such as 1–ethyl–3–methyl–imidazolium trifluorochloroborate ([EMIM][BF_3_Cl]) IL, can interact with CO_2_ through a Lewis base adduct, making them active for CO_2_ electroreduction reaction [54]. Ionic Liquids (ILs) are highly efficient materials for CO_2_ electroreduction, as they serve as both electrolytes and functionalized materials. They offer kinetic effects that reduce the energy required for intermediate ${\mathrm{CO}}_{2}^{-}\cdot$ formation, resulting in an improved CO_2_ ERR efficiency. In one study, methylimidazolium groups are attached to the edge of an iron porphyrin, to create a pre-arranged environment that displays outstanding selectivity for CO production at low overpotentials when using water as the solvent and proton source [5]. According to the findings of Vianney O. Santos Jr. et al., who conducts an investigation on the electrochemical reduction in CO_2_ on a copper electrode in BMI·BF_4_ using SERS as a monitoring tool, the peaks at 1532, 1492, 1281, 1091, and 672 cm^−1^ are assigned to the vibrational modes of the BMI–CO_2_ adduct [29]. Based on the spectra and previous research, it seems that the peaks at 1100, 1457, and 1509 cm^−1^ can also be attributed to the vibrational modes of the BMI-CO_2_ adduct. This is consistent with previous studies on the electrochemical reduction in CO_2_ in ionic liquids, which have shown that ILs can interact with CO_2_ to form adducts [51]. It is interesting to note that the SERS spectra do not show any signal that could be attributed to the ${\mathrm{CO}}_{2}^{-}\cdot$ anion radical, which has been observed in other studies using pulse radiolysis time-resolved resonance Raman spectroscopy and DFT calculation [55]. The peaks indicate the oxidation state of cobalt at 1347 and 1390 cm^−1^ suggest the presence of both Co^II^ and Co^III^, with the intensity of the peak for Co^II^ as the potential becomes more negative due to the reduction in Co^III^. The vibration modes correspond to the υ(C-C) stretch vibrations at 1564 cm^−1^ suggest that the oxidation state of cobalt is mainly Co^III^ with a small vibration mode at 1534 cm^−1^ attributed to Co^II^ and overlapping with the BMI·CO_2_ feature.

The feature at 2124 cm^−1^ is assigned to the *υ*_1_ CO stretching modes, where CO adsorbed on the cobalt center as shown in Figure 6a. The IR spectra of Fe(TPP) and Co(TPP) adducts with CO are discussed in literatures, showing CO stretching bands at 2138 and 2149 cm^−1^, which are attribute to matrix isolated monomeric carbon monoxide and CO bound H_2_O traces, respectively [56,57]. The bands at 1974 and 2036 cm^−1^ correspond to the *υ*(CO) of mono– and bis–carbonyl complex of Fe(TPP), while a single band at 2078 cm^−1^ is assigned to the bis–carbonyl complex Co(TPP)(CO)_2_ [58,59]. Therefore, it is reasonable to assign the peal at 2124 cm^−1^ to the CO stretching band originating from monomeric CO. The absence of the *υ*(CO) band for the bis–carbonyl may be because the fact that the Co porphyrin in the study is coordinated to the Ag electrode, resulting in a five-coordinate complex that provides insufficient coordination positions for the bis–carbonyl group.

According to Figure 6b,c, the SERS signals at 2124 cm^−1^ observed in the study shifted to lower wavenumbers as the potential is made more negative. The changes in peak positions and intensity with applied potentials are displayed. The *ν*_CO_ peak in * CO exhibits the electrochemical Stark effect, characterized by a Stark tuning rate of 21 ± 1.4 cm^−1^/V (Figure 4b), which is a typical observation for CO chemisorbed on atop sites. The red shift (positive rate) indicates that the C≡O dipole is elongated by the negative electric field that is directed towards the surface. The dependence of intensity on the potential indicates that with the negative electric field CO is kicked-off from the surface of the electrode surface.

### 2.4. Mechanism of CO_2_ Electrochemical Reduction

Drawing from the preceding discussion, a plausible mechanism for the electrochemical reduction in CO_2_ can be posited as in Figure 1. The mechanism initiates with the reduction in the metal center of the porphyrin, resulting in the formation of [PP–Co^I^]^-^ anion, which has a partially reduced metal center and ligand. The electron transfer takes place in mixed molecular orbitals from the metal ion and the ligand. Subsequent to the ligand reduction, electron transfer leads to the protonation of the anion, generating the pholorin anion [H–PP–Co^I^]^−^. The ionic liquid (IL) interacts with CO_2_ via a Lewis base adduct, resulting in the BMI·CO_2_ adduct, which becomes activated to bind to the cobalt center, forming the intermediate [H–PP–Co^I^–CO_2_]^−^. The intermediate [H–PP–Co^I^–CO_2_]^−^ undergoes intermolecular chemical bond recombination, leading to the formation of [PP-Co^I^-COOH]^−^ intermediate. Through the protonation of the -OH group of the intermediate and the subsequent loss of a molecule of H_2_O, a CO-adsorbed intermediate [PP–Co^I^–CO]^−^ is generated, which eventually releases CO as the final product.

## 3. Materials and Methods

### 3.1. Chemicals

The chemicals and materials used in the study include 30% H_2_O_2_ and AR grade concentrated H_2_SO_4_, which are purchased from Sigma-Aldrich (Shanghai, China). Cobalt protoporphyrin was obtained from Frontier Scientific, USA. An IL, specifically (1–Butyl–3–methylimidazolium tetrafluoroborate, BMI·BF_4_), was used without further purification. Water was prepared using Millipore water (18.2 MΩ).

### 3.2. Preparation of Working Electrode

A conventional Au disk electrode with a diameter of 0.2 mm was first polished with alumina slurry, using 0.1 μm and 0.05 μm diameter particles sequentially and then sonicated in water for 10 min. Nanostructured Ag particles with a size of 140 nm were synthesized following the method described in literature [60]. Next, 10 μL of the Ag nanoparticles solution was dropped onto the conventional Au disk electrode for twice, which was then dried under vacuum. 0.5 mM solution of cobalt protoporphyrin was prepared in ethanol. Then, 10 μL of the solution was dropped onto the dried nanostructured Ag coated electrode and dried again under vacuum. For the electrochemical measurements, the L-shape carbon paper electrode was dipped into 0.5 mM cobalt protoporphyrin solution in ethanol for 10 min, then took out and dried in air.

### 3.3. Electrochemical Measurements

The electrochemical measurements were performed using a potentiostat (CHI 660e) and a home-made conventional three-electrode configuration. The working electrode was the cobalt protoporphyrin immobilized Ag electrode as working electrode, while the counter electrode was a Pt wire. The reference electrode used was an Ag/Ag^+^ reference electrode that was prepared according to literature [61]. Prior to the experiments, all glassware was immersed in piranha solution (3:1 concentrated H_2_SO_4_/30% H_2_O_2_), washed with deionized water and dried with N_2_. The supporting electrolyte used was a 25 mL ionic liquid solution. Additionally, the influence of water on the electrochemical reduction in CO_2_ was investigated using supporting electrolyte consisting of a mixture of IL solution and water in a 9:1 volume ration. The electrochemical measurements have been conducted in a three-electrode H-cell as employed with a L-shaped carbon paper (active size is 1 × 1 cm) as the working electrode, platinum foil (2 × 2 cm) as the counter electrode, and an Ag/Ag^+^ reference electrode. The linear sweep voltammetry is conducted with a scan rate of 0.1 V/s in a potential range of 0 V to −1.5 V (vs. Ag/Ag^+^). The cyclic voltammetry (CV) is conducted in a potential range of 0 V to −2.0 V (vs. Ag/Ag^+^) with other parameters same as LSV.

### 3.4. SERS Measurements

The in situ-electrochemical SERS spectra were collected using a confocal Raman system called WITec Alpha 300R. The laser illumination and signal collection were performed using a water immersion objective with a numerical aperture of 1.0 and magnification of 60. A He–Ne laser was used to generate the 632.8 nm excitation with a power of 25 μW. The laser power was measured at the entrance of the microscope objective. The integration time for the measurements was 10 s.

## 4. Conclusions

In conclusion, this study investigated the molecular mechanism of cobalt porphyrin catalyzed CO_2_ electrochemical reduction using in-situ SERS. The results showed that cobalt porphyrin adsorbs successfully on the surface of the Ag electrode, exhibiting marker vibration modes of *υ*(C–N) and *υ*(C–C) at 1351, 1399 cm^−1^ and 1532, 1558 cm^−1^ for Co^III^ and Co^II^, respectively. Water served as a proton source, and the vibration modes assigned for the out-of-plane mode of protonated porphyrin molecule indicated that the key intermediate for competing HER and CO_2_ electrochemical reduction is pholorin. The use of ionic liquids helps to activate CO_2_ through the formation of the BMI·CO_2_ adduct, which leads to the formation of [PP–Co–COOH]^−^ intermediate for the final product. These findings provide insights into the molecular-level understanding of CO_2_ electrochemical reduction and can aid in the design of more efficient and environmentally sustainable CO_2_ conversion systems. However, the molecular structure of the cation in ionic liquids plays a critical role in determining the electrochemical reduction in CO_2_ reaction. The properties of the cation, such as ion size, shape, charge and functional groups can significantly influence the reaction mechanism, kinetics, and selectivity. Further research is needed to explore the relationship between the cation structure and electrochemical performance and to develop new strategies for optimizing the catalytic properties of ionic liquids.

## Data Availability

Not applicable.

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
