# Peer review of "Investigation of Molecular Mechanism of Cobalt Porphyrin Catalyzed CO2 Electrochemical Reduction in Ionic Liquid by In-Situ SERS"

_molecules, 2023, doi:10.3390/molecules28062747_

Round 1
Reviewer 1 Report
The manuscript entitled “Investigation of Molecular mechanism of cobalt porphyrin catalyzed CO2 electrochemical reduction in ionic liquid by in-situ SERS”, which Authors are: Feng Wu, Fengshuo Jiang, Jiahao Yang, Weiyan Dai, Donghui Lan, Jing Shen, Zhengjun Fang, explored the electrochemical reduction of CO2 using room temperature ionic liquids as solvents or electrolytes. They investigated the electrochemical reduction of CO2 using a cobalt porphyrin molecular immobilized electrode in 1-n-butyl-3-methyl imidazolium tetrafluoroborate (BMI.BF4) room temperature ionic liquids, through in-situ surface-enhanced Raman spectroscopy (SERS).
The topic is of special interest in the minimization of the environmental impact of CO2 emissions but some issues must be clarified before the manuscript is published.
1) Authors carried out a qualitative study of the electrochemical reduction of CO2. It would be very important to present quantitative results, in terms of the substances produced from this reduction.
2) It would also be important to study in this work the influence of the cation of the ionic liquid
Reviewer 2 Report
1) The paper need an extended bibliographic background, considering the following basic search:
https://www.base-search.net/Search/Results?lookfor=electrochemical+CO2+technique+reduction&name=&oaboost=1&newsearch=1&refid=dcbases
2) Author must refer and compare their study against: Kang, J. E., Palgunadi, J., Kim, H., Cheong, M., & Kim, H. S. (2009). Separation of CO2 using room temperature ionic liquids. In American Chemical Society - 238th National Meeting and Exposition, ACS 2009, Abstracts of Scientific Papers (ACS National Meeting Book of Abstracts).
3) The paper could be published if the authors emphasize the originality of their research, compared to previously reported research.
Round 2
Reviewer 2 Report
The authors attended to the observations prior to the manuscript, which is why it seems suitable for publication.